# Automated Approach to In Vitro Image-Guided Photothermal Therapy with Top-Down and Bottom-Up-Synthesized Graphene Quantum Dots

**DOI:** 10.3390/nano13050805

**Published:** 2023-02-22

**Authors:** Bong Lee, Gretel A. Stokes, Alina Valimukhametova, Steven Nguyen, Roberto Gonzalez-Rodriguez, Adam Bhaloo, Jeffery Coffer, Anton V. Naumov

**Affiliations:** 1Department of Physics and Astronomy, Texas Christian University, Fort Worth, TX 76109, USA; 2Department of Physics, University of North Texas, Denton, TX 76203, USA; 3Department of Chemistry and Biochemistry, Texas Christian University, Fort Worth, TX 76109, USA

**Keywords:** photothermal therapy, graphene quantum dots, fluorescence imaging, near-infrared fluorescence

## Abstract

Graphene-based materials have been the subject of interest for photothermal therapy due to their high light-to-heat conversion efficiency. Based on recent studies, graphene quantum dots (GQDs) are expected to possess advantageous photothermal properties and facilitate fluorescence image-tracking in the visible and near-infrared (NIR), while surpassing other graphene-based materials in their biocompatibility. Several GQD structures including reduced graphene quantum dots (RGQDs) derived from reduced graphene oxide via top-down oxidation and hyaluronic acid graphene quantum dots (HGQDs) hydrothermally bottom-up synthesized from molecular hyaluronic acid were employed to test these capabilities in the present work. These GQDs possess substantial NIR absorption and fluorescence throughout the visible and NIR beneficial for in vivo imaging while being biocompatible at up to 1.7 mg/mL concentrations. In aqueous suspensions, RGQDs and HGQDs irradiated with a low power (0.9 W/cm^2^) 808 nm NIR laser facilitate a temperature increase up to 47.0 °C, which is sufficient for cancer tumor ablation. In vitro photothermal experiments sampling multiple conditions directly in the 96-well plate were performed using an automated simultaneous irradiation/measurement system developed on the basis of a 3D printer. In this study, HGQDs and RGQDs facilitated the heating of HeLa cancer cells up to 54.5 °C, leading to the drastic inhibition of cell viability from over 80% down to 22.9%. GQD’s fluorescence in the visible and NIR traces their successful internalization into HeLa cells maximized at 20 h suggesting both extracellular and intracellular photothermal treatment capabilities. The combination of the photothermal and imaging modalities tested in vitro makes the GQDs developed in this work prospective agents for cancer theragnostics.

## 1. Introduction

Cancer remains one of the deadliest diseases, leading to almost 10 million deaths each year worldwide [1,2]. Current cancer treatment options such as surgery, chemotherapy, and radiation therapy possess side effects that affect both cancer and healthy tissue [3,4,5]. Cancer-specific treatment options that may lead to improved patient care and outcomes are now becoming a focus of active scientific inquiry. One of the promising approaches, photothermal therapy (PTT), can effectively eradicate tumors when localized specifically to the cancer site, reducing off-site side effects [3,6]. PTT utilizes highly absorbing materials termed photothermal agents, which uptake energy from an irradiation source and dissipate it in the form of heat to trigger cell death [3]. Cancer cells are less resistant to elevated temperatures compared to the healthy cells due to their compromised DNA repair mechanisms [7]. Cancer cell death is triggered by PTT at tissue temperatures above 42 °C, which leads to DNA damage and protein denaturation [8,9,10]. First, the irreversible tissue damage occurs at 42 °C, [10] while necroptosis and apoptosis are initiated at 46 °C [11] and fully achieved at 49 °C [12]. To minimize laser ablation of the adjacent healthy tissue, cancer tumors can be saturated with photothermal agents highly absorbing in the near-infrared (NIR). NIR presents two biologically safe irradiation wavelength regions at 650–950 nm (first biological window) and 1000–1350 nm (second biological window) [13]. Within these biological windows, tissue scattering and absorption are minimized, which allows NIR light to penetrate a few centimeters into the tissue [13,14,15] to be only absorbed by the photothermal agents. Nanoscale PTT agents can be preferentially targeted to tumor sites by specific receptor-mediated approaches. Nanomaterials within a size range of up to 200 nm [16] also accumulate in tumors passively due to the inherent leaky and disorganized tumor vasculature [10] via the enhanced permeability and retention (EPR) effect [17,18,19].

Current PTT research focuses on noble metal and semiconductor nanoparticles rather than conventional dyes due to the nanosized materials’ high NIR absorption and photostability [20,21,22,23,24,25]. Noble metal nanoparticles achieve PTT via the localized surface plasmon resonance when optically excited plasmon undergoes nonradiative decay via intraband and interband transitions channeling energy to heat [26,27]. In semiconductor nanoparticles, excitons generated by visible and near-infrared irradiation can experience nonradiative decay and dissipate their energy in the form of heat [26]. However, noble metal structures such as gold nanoparticles are limited in their long-term in vivo use due to their non-biodegradability [28], while the use of semiconductor nanomaterials is often restricted by their toxicity [29].

Graphene and its derivatives, such as single-walled carbon nanotubes (SWCNTs) and graphene oxide (GO), have attracted significant attention as alternative photothermal agent candidates [30] due to their advantageous properties, especially their high absorption throughout the visible and even NIR [31,32,33,34]. Their high photothermal conversion is attributed to substantial optical absorption in the NIR arising from their low dimensionality-based electronic structure and low-energy vibrational modes [20,35,36,37,38]. As an example, GO is known to exhibit successful photothermal conversion tested in vitro. Functionalized with porphyrin, GO can facilitate the ablation of glioblastoma cells (U87-MG), decreasing their viability down to 24% after 10 min of 808 nm irradiation at 2.5 W/cm^2^ [39]. Such laser power enables the heating of GO dispersed in cell media at 200 µg/mL from 24 °C to ~60 °C sufficient for rapid tumor ablation [39]. To enhance the effects of PTT, different therapeutics can be conjugated to GO, facilitating synergistic cancer treatment via complementary chemotherapy, [30] gene therapy, [40] or photodynamic therapy [41]. Another graphitic nanomaterial, SWCNTs, due to their intrinsic NIR fluorescence, can perform NIR image-guided PTT in vivo [42]. PEGylated SWCNTs successfully demonstrated photothermal treatment capabilities in xenograft tumor mouse models [35] and cancer cells that metastasized from the primary tumor site [42]. Conjugating SWCNTs with folic acid to target folate receptors overexpressed in cancer cells allows for their preferential accumulation and focused PTT at the cancer sites [24]. While showing high efficacy treatment and complementary diagnostic imaging capabilities, graphene-based derivatives, including SWCNTs and GO, are still limited in their translation to the clinic mainly due to the reports of potential off-target toxicity, and/or immunogenicity exacerbated by long-term tissue accumulation [43,44]. Efforts to address these issues have led to the development of biocompatible graphene derivatives such as graphene quantum dots (GQDs). These zero-dimensional nanoparticles exhibit size-dependent visible fluorescence, attributed to quantum confinement effects. Select GQD structures can also absorb and emit in the NIR, which arises from defect states and/or dopants [43,44,45]. Another benefit of the GQDs is their scalable and cost-efficient synthesis that can be performed without the use of toxic precursors [46,47]. There are two approaches to synthesizing GQDs: top-down and bottom-up. For top-down synthesis, bulk carbon materials such as coal, graphite, carbon fibers, and graphene and its derivatives undergo physical or chemical scission processes to generate nanometer-sized GQDs [48,49]. Among those, reduced graphene oxide (RGO) presents an advantageous micro- to nano-scale precursor alternative that is widely available due to its direct production from GO, which, in turn, can be synthesized from graphite [50,51]. Our recent works show the possibility of deriving GQDs from RGO via a one-step oxidation procedure employing sodium hypochlorite as a facilitator of oxidative scission [44]. As an alternative, the bottom-up synthetic approach starting with carbon-containing molecules undergoing dehydration and polymerization in hydrothermal and/or solvothermal reactions also produces nanometer-sized GQDs. The source of these carbonaceous molecules is not limited to isolated organic compounds [45,52,53,54] but can also include biomass [55,56,57,58,59]. Our previous works demonstrate the successful synthesis of bottom-up GQDs using a variety of molecular precursors including biocompatible glucosamine [60]. This process yields amphiphilic non-toxic GQDs at over 1 mg/mL concentrations that are biodegradable in cellular environments. Due to their simple synthesis and apparent biocompatibility, GQDs prove to be prospective candidates for biosensing, bioimaging, and therapeutic delivery [47,60,61]. Making GQDs also capable of photothermal treatment can enhance their capacity as standalone cancer therapeutic agents and/or facilitate irradiation-stimulated drug release from the GQD platform. The presence of substantial absorption and emission in the NIR for some GQD structures suggests the possibility of using those for both NIR imaging and PTT applications [62,63,64,65]. NIR fluorescence can provide real-time feedback on the PTT agent’s location and help focus the irradiation onto the nanomaterial’s localization area. This work aims to assess the benefits of using either top-down or bottom-up synthesized GQDs as dual PTT and bioimaging agents in vitro.

While a variety of molecular precursors are available for bottom-up synthesis, non-toxic precursors are preferable to facilitate the high biocompatibility of all synthetic products. The synthetic approach chosen in this work utilizes an acidic mucopolysaccharide, hyaluronic acid (HA), which acts as one of the main components of the extracellular matrix [54,66]. In addition to being a convenient source of carbon and oxygen, the HA backbone exhibits tumor-targeting capabilities from its specific binding to the CD-44 cell surface adhesion receptors highly expressed in many cancer cell types for their dissemination and metastasis [67,68,69]. Thus, targeting the CD-44 receptor can enhance nanomaterial’s tumor accumulation in vivo [70,71]. HA-derived carbon dots which, unlike GQDs, do not possess graphitic structure have already been previously synthesized using a bottom-up approach and retained HA targeting properties [54,72]. Thus, there is a promise that HA-synthesized GQDs (HGQDs) may also possess targeting capabilities and facilitate cancer-focused PTT. These advantages explain the choice of HA as a molecular precursor for GQD synthesis. Additionally, the availability and low cost of HA make the HGQD synthesis scalable. The top-down GQDs (RGQDs) are synthesized via the aforementioned scission approach from an RGO precursor. In addition to similar low cost and synthetic scalability, these GQD platforms are known to exhibit high-yield near-infrared fluorescence enabling both treatment and diagnostic capabilities. Therefore, as compared to other GQDs previously tested for PTT applications, the ones developed in this work concomitantly offer visible and NIR image-tracking and possess up to 10-fold higher biocompatibilities (Appendix A). These nanomaterials can pave the way for the development of clinically relevant multimodal cancer theragnostic agents.

Finally, to evaluate the PTT capabilities of HGQDs and RGQDs in vitro, a novel cost-effective automated approach to larger-scale PTT studies is introduced. Automated equipment is widely used in commercial and research settings to reduce labor-intensive tasks, increase throughput, and minimize human error [73,74]. However, commercial instruments often perform only select functions and may come at a prohibitively high cost [74]. Currently, there are no instruments directly allowing for the analysis of nanomaterials’ PTT effect in vitro. Such systems, however, can be developed in-house by retrofitting a 3D printer for PTT applications. Due to their low cost, 3D printers are available to the general population, democratizing the tool for common use in personal and laboratory settings [75,76,77,78]. Both the software and hardware components of a 3D printer are retrofittable for the users’ applications. In this work, we adapted a 3D printer to design a novel in vitro PTT testing system performing 96-well plate experiments and allowing for precise individual well laser irradiation and temperature measurements to assess the effects of PTT. Moreover, a consecutive MTT assay in the same well plate can provide the evaluation of cell viability post-treatment. This automated high-throughput technique developed, for the first time, in this work can streamline PTT in vitro assessment for a variety of therapeutics allowing to select from a plethora of nanoparticle configurations and treatment conditions.

## 2. Experimental Section

### 2.1. Synthesis of Top-Down Graphene Quantum Dots

RGQDs were synthesized via a top-down approach by oxidizing high porosity reduced graphene oxide (HP-RGO-025G, Graphene Supermarket, Ronkonkoma, NY, USA). The reaction was carried out as follows: RGO was dispersed in DI water at a concentration of 0.25 mg/mL and 1 mL of sodium hypochlorite (LC246364, LabChem, Zelienople, PA, USA) was added to the aqueous RGO suspension. The vials were shaken, capped, and allowed to react for 2 days. The products were further purified using a 1 kDa dialysis bag (132104, Repligen, Waltham, MA, USA) in DI water for 24 h, where the water was changed every 30 min during the first 3 h. Finally, the purified suspension was passed through a 0.22 µm syringe filter for filtration of the unreacted RGO precursor and sterilization purposes.

### 2.2. Synthesis of Bottom-Up Graphene Quantum Dots

A powdered sample of 5 kDa hyaluronic acid (HA5K-5, Lifecore Biomedical, Chaska, MN, USA) was heated up to 300 °C for 5 min on a hot plate. After cooling to room temperature, the product was dispersed in water via bath ultrasonic treatment and centrifuged to isolate hydrophobic precipitates. The collected supernatant was further centrifuged in a 10 kDa centrifugation filter (10 kDa, HighSpeed™ Microcentrifuge, Southwest Science, Roebling, NJ, USA) to remove unreacted hyaluronic acid leaving larger GQD structures. Finally, the sample was passed through a 0.22 µm syringe filter for additional purification and sterilization purposes.

### 2.3. Characterization

Transmission electron microscopy (TEM, JEOL JEM-2100, Peabody, MA, USA) with energy dispersive x-ray spectroscopy (EDS, JEOL, Peabody, MA, USA) was used to assess GQD size, crystallinity, and composition. TEM sample was prepared by air-drying GQD suspensions on carbon-coated 200 mesh copper grids (Electron Microscopy Sciences, CF300-CU, Radnor, PA, USA). UV-Vis absorption spectra of all samples were acquired using Cary 60 (Agilent Technologies, Santa Clara, CA, USA) spectrophotometer in the range of 200 to 1100 nm. Visible fluorescence spectra were measured using a NanoLog (HORIBA Scientific, Piscataway, NJ, USA) spectrofluorometer, while NIR fluorescence spectra were assessed with AvaSpec-HS-TEC spectrofluorometer (Avantes, Apeldoorn, The Netherlands) with a 0.9 W/cm^2^ 808 nm laser (808MD-12V-BL, Q-BAIHE, China) excitation.

### 2.4. Photothermal Effect in Solution

The photothermal effect was characterized as follows: separately, 3 mL of aqueous RGQDs at 1.5 mg/mL and 3 mL of HGQDs at 1.7 mg/mL were subjected to a 0.9 W/cm^2^ 808 nm laser irradiation for 45 min. DI water was irradiated as a control. The temperature of each solution was continuously monitored by the submerged thermocouple. The experiments were performed in an oven held at a biological temperature of 37 °C with suspensions being magnetically stirred to maintain a constant temperature throughout the sample.

### 2.5. Cell Culture

HeLa cells were utilized for in vitro studies. These cells were cultured using a complete medium consisting of DMEM (D6046, Sigma-Aldrich, St. Louis, MO, USA) supplemented with 10% FBS (16140-063, Gibco, Dublin, Ireland), L-Glutamine (G7513, Sigma-Aldrich, St. Louis, MO, USA), non-essential amino acid solution (M7145, Sigma-Aldrich, St. Louis, MO, USA), and 1% penicillin/streptomycin (P4333, Sigma-Aldrich, St. Louis, MO, USA). The cell culture was held in an incubator at 37 °C with 5% CO_2_ (Midi CO_2_, Thermo Fischer Scientific, Waltham, MA, USA).

### 2.6. Cell Viability Assay

To assess the biocompatibility of the RGQDs and HGQDs, MTT (3-(4–dimethylthiazol-2-yl)-2,5 diphenyltetrazolium bromide) assay (Invitrogen M6494, Thermo Fischer Scientific, Waltham, MA, USA) was used. A total of 5000 cells per 200 µL were seeded into each well in a 96-well plate (701001, Nest Scientific, Wuxi, China) and kept in an incubator. RGQDs at 1.5 mg/mL and HGQDs at 1.7 mg/mL were added into different wells in two-fold serial dilutions in DMEM. After 24 h of post-treatment incubation, the complete medium in the 96-well plate was replaced with 100 µL of 1 mg/mL of MTT and stored in the incubator for 4 h. This step allowed for the metabolization of the MTT that was further replaced with 100 µL of DMSO. Finally, the 96-well plate was placed on an orbital shaker for 5 min, which was followed by the absorption measurement at 570 nm for each experimental well in an absorption plate reader (μQuant, BioTek Instruments, Winooski, VT, USA). These measurements indicated the amount of formazan generated via the reduction in the MTT by NAD(P)H-dependent cellular oxidoreductase enzymes of the viable cells, which was, therefore, proportional to the number of viable cells.

### 2.7. Automated Laser Irradiation and Temperature Measurement

A 3D printer (CR-10mini, Creality, Shenzhen, China) was retrofitted to be equipped with a 0.9 W/cm^2^ 808 nm laser (808MD-12V-BL, Q-BAIHE, China) and a T-type thermocouple (TL0024, Perfect Prime, New York, NY, USA). They were attached to the extruder component of the 3D printer. The retrofitted 3D printer was placed inside a biosafety hood (Forma 1284, Thermo Fisher Scientific, Waltham, MA, USA) with a DI water reservoir to maintain the humidity of the biosafety hood environment. An iterative code was written in G-code (Appendix A) to set the temperature of the heat bed of the 3D printer to biological 37 °C, position the laser and thermocouple at individual wells, and turn the laser on or off. A wireless data logger thermometer software (TC0521, Perfect Prime, New York, NY, USA) was used to collect and tabulate temperature measurements acquired by the thermocouple. The ultimate function of such retrofitted 3D printer device depicted in the schematic in Appendix A involves irradiating the wells of the 96-well plate for 0, 1, 5, and 10 min with subsequent thermocouple well temperature measurements. The post-treatment cell viability of HeLa cells laser-treated in the 96-well plate was analyzed via the MTT assay.

### 2.8. Confocal Fluorescence Microscopy Imaging

Confocal fluorescence microscopy imaging was performed with a semi-motorized inverted microscope (IX73P2F, Olympus, Center Valley, PA, USA) using an IR-corrected UPLANAPO 60x/0.90na objective (1-UB831, Olympus, Center Valley, PA, USA) coupled to visible and near-infrared imaging setups. The visible fluorescence setup utilized a CMOS (Prime 95B, Photometrics, Tucson, AZ, USA) camera and a confocal disk-spinning unit (Olympus, Center Valley, PA, USA) with a 460 ± 20 nm filtered lamp excitation and 535 ± 20 nm filtered emission used in this work. The NIR setup involved an InGaAs FPA (ZephIR^TM^ 1.7, Photon etc., Montreal, Quebec, Canada) camera and a hyperspectral imager for spectrally resolved image collection. The NIR emission was excited with an 808 nm laser and collected at the corresponding GQD emission range within 900–1200 nm. 

## 3. Results and Discussion

In this work, GQDs were synthesized via the top-down and bottom-up approaches. RGQDs were produced from commercially available RGO in the top-down approach. The reaction was catalyzed by sodium hypochlorite (NaOCl) photodissociation, generating highly reactive oxygen free radicals to oxidize the graphitic structure [44,79]. In the bottom-up approach, HGQDs were synthesized via the dehydration reaction of the HA, during which polymeric aromatic structures were formed [45]. Size distributions of RGQDs and HGQDs assessed via TEM imaging averaged at 9.6 ± 1.0 and 5.0 ± 0.3 nm, respectively (Figure 1), indicating that nanoscale particles were created as a result of both top-down and bottom-up reactions. In vivo biodistribution studies of GQDs and carbon dots of similar sizes reported their accumulation mainly in the liver, spleen, and kidneys, where they later underwent rapid renal clearance reducing potential off-target effects, while still being capable of some EPR-based passive tumor targeting [44,65]. Some aggregates present in TEM images and reflected in GQD size distributions could have formed during the air drying of GQD suspensions on the TEM grid. In suspension, however, both GQDs showed high colloidal stability for over 6 months with substantially negative Zeta potentials of −18.7 ± 2.2 and −49.4 ± 0.4 mV for HGQDs and RGQDs, respectively (Appendix A).

EDS-based elemental analyses showed that HGQDs possessed more carbon (C: 98.1%) and less oxygen (O: 1.9%) when compared to RGQDs (C: 91.3%, O: 8.7%) (Appendix A). The presence of oxygen functional groups in the GQD structure made them amenable to chemical functionalization [44,60,61]. Furthermore, HRTEM and FFT image analyses indicated that both RGQDs and HGQDs were crystalline in structure with an interplanar distance of 0.21 nm corresponding to the lattice spacing of (100) graphene plane [80]. This finding suggests the possibility of non-covalent attachment of hydrophobic therapeutic cargo to the GQDs [47] and shows the potential for high photothermal conversion efficiency demonstrated previously for graphene and its derivatives [36,37,38]. Absorption measurements (Figure 2) indicated that both GQDs exhibited characteristic graphitic $\pi \text{-}\pi^{*}$ transitions along with an absorption tail extended into the longer wavelength region. While being minor when compared to the UV absorption, this tail, in part attributed to surface defect states and sub-gap transitions, still provides an avenue for photothermal conversion. Comparing both absorption spectra showed a narrower blue-shifted $\pi \text{-}\pi^{*}$ transition peak for RGQDs, while HGQDs possessed broader red-shifted absorption with a more extended long wavelength tail (Figure 2a).

Oxygen functional groups expected to contribute to the absorption shoulder in the 300 nm region facilitate sufficient water-solubility for RGQDs and HGQDs at the biologically relevant concentrations. The presence of oxygen functional groups was further validated by Fourier transform infrared spectroscopy (FTIR) measurements (Appendix A). Both GQDs showed stretching vibrations from O–H, C–H, and C=O/C=C bonds with peaks at 3380, 2909, and 1587 ${\mathrm{cm}}^{-1}$ for RGQDs and 3316, 2925, and 1597$\;{\mathrm{cm}}^{-1}$ for HGQDs. Unlike RGQDs, HGQDs exhibited an additional stretching vibration for the C–O bond at 1038 ${\mathrm{cm}}^{-1}$. Both bottom-up and top-down approaches yielded GQDs fluorescing in the visible and NIR (Figure 2b). Their broad visible emission (Figure 2c,d) can arise from a distribution of GQD sizes in the sample. GQDs are small enough to serve as regions of electronic confinement, with each GQD size possessing a different band gap and a set of quantum confinement-dictated excitation and emission wavelengths [81]. Localized electronic defects in the GQD graphitic structure originating from oxidation and oxygen functional groups can lead to excitation-independent NIR emission. With an 808 nm laser excitation, both HGQDs and RGQDs exhibited NIR fluorescence spanning the wavelength range of 850 to 1100 nm with a peak at ~900 nm (Figure 2b). This fluorescence occurred within the biological absorption windows, which is desirable for facilitating NIR image-guided photothermal applications. The similarity of the spectral features of both GQD types can be explained by the presence of similar functional groups (Appendix A).

In order to assess maximum non-toxic concentrations for PTT applications, GQD biocompatibility was investigated in HeLa cancer cells intended as a biological test platform for the photothermal therapeutic capabilities of the GQDs. As shown by the MTT assay, RGQDs and HGQDs yielded over 80% cell viability at concentrations of up to 1.5 and 1.7 mg/mL, respectively, appearing non-toxic at high doses (Figure 3), which makes them relevant for further clinical applications. NIR imaging facilitated by RGQDs in vivo also gives promise for their clinical translation [44]. Therefore, given significant biocompatibility along with some absorption in the NIR, GQDs can serve as potential candidates to be tested for PTT applications.

Photothermal conversion efficiency of RGQDs and HGQDs was first assessed at biocompatible concentrations in an aqueous suspension. The photothermal studies reported herein utilized an 808 nm laser excitation, since the absorption coefficient of water at that wavelength is minimized [82]. Thus, unlike visible wavelengths, an 808 nm excitation in the first water window allows for deeper tissue imaging, making this excitation method potentially translatable to in vivo applications. In contrast to several previous works [24,38,65,83], the photothermal effect was assessed here at a biologically relevant baseline temperature of 37 °C in a temperature-controlled oven simulating a large heat sink provided by the human body. Upon irradiation, the HGQD and RGQD suspensions experienced respective temperature increases of up to 46 and 47 °C, while a water control sample only heated up to 37.3 °C (Appendix A). These temperatures, which are sufficient to initiate necrosis and apoptosis in cancer cells, exceed the ones reported in PTT studies with a number of nanomaterials [84,85,86]. To ensure their stability as photothermal agents, GQD samples had to undergo three cycles of laser irradiation, reaching similar temperatures averaged and reported with appropriate error ranges (Figure 4). It is worth noting that almost a maximum heating was achieved already at 10–15 min of irradiation, after which solution temperatures reached nearly an asymptotic limit. Thus, shorter time frames can be used for cancer photothermal therapy reducing potential detrimental effects on healthy tissue.

In order to test the photothermal treatment capability of the GQDs in vitro, a unique automated 3D printer-based system was developed and utilized to directly irradiate cells and measure cell culture temperatures in a 96-well plate. HeLa cells treated with RGQDs at 1.5 mg/mL and HGQDs at 1.7 mg/mL for 24 h served as a cancer cell culture model in the automated PTT treatment. This system can perform continuous in vitro experiments during which the laser is programmed via an open-source G-code software RS-274 (Appendix A) to irradiate a sequence of wells with controlled time and power settings. After the irradiation of each well, the add-on thermocouple of the system measured well temperature and recorded it in the file. In order to ensure minimal cross-contamination between the wells, additional wells filled with DMEM were used to rinse and clean the thermocouple probe by dipping it into the well repeatedly. These wells also acted as cooling sinks for the temperature probe ensuring that it entered each new experimental well at the same initial temperature and, thus, did not perturb the measurement. The heat bed of the 3D printer was fitted to the 96-well plate, providing direct thermal contact to keep the base temperature at a biological 37 °C. This feature, which was rarely present in other in vitro PTT studies, simulates the massive 37 °C base temperature heat reservoir provided by the human body. Such experimental design, as well as proper calibration, enabled accurate and reproducible measurements. The temperature calibration of this instrument involved successfully measuring 37 °C in all wells that will be utilized in the experiment. In order to accomplish that, only select wells closer to the middle of the plate were used. Each experimental well was surrounded by empty wells to facilitate its thermal insulation (Appendix A). To prevent excessive evaporation of solution from the wells and maintain sterility, the experiment was performed in a biosafety hood with a large water reservoir humidifying the environment. After the irradiation and temperature measurements, the same 96-well plate was further analyzed via the MTT assay to evaluate treatment-derived effects on cell viability. All experimental conditions were tested in this high-throughput approach in a single 96-well plate. This included non-treatment control wells as well as those treated with each type of GQD and irradiated for 1, 5, and 10 min.

Temperature measurements help assess the efficacy of PTT for a particular condition. For instance, HeLa control wells irradiated for a maximum time of 10 min exhibit temperature increased only up to 38.9 ± 0.5 °C with no effect on cell viability (Figure 5a). This control test showed that the heating was negligible without the addition of the photothermal agents. In contrast, a 10 min irradiation of HeLa cells treated with RGQDs increased well temperatures to 43.5 ± 0.1 °C and dropped cell viability down to 38.2 ± 4.3%. HGQDs-treated cells upon 10 min irradiation experienced an even more drastic cell viability decrease down to 22.9 ± 1.8% due to PTT heating up to 54.5 ± 0.3 °C (Figure 5b). Again, such heating surpassed PTT temperatures achieved with multiple nanomaterial platforms [87,88,89]. The table of temperatures and cell viabilities delineates these effects per condition (Appendix A), showing the reproducibility of achieved anticancer treatment. The temperatures in the range of 46 to 55 °C were reported to cause irreversible cellular damage and cell death [90], which validates the observed low cell viabilities within the treated wells. Having two assessment factors (well temperature and cell viability) allows for highly deterministic analysis of the nanomaterials’ PTT capabilities. Compared to the present work, previous studies utilizing GQDs for PTT applications (Appendix A) lack to quantify these critical assessment factors. As the observed heating can originate from GQDs located both intracellularly and extracellularly, we verified their intracellular translocation and the capability for in vitro tracking via both visible and NIR fluorescence microscopies.

HeLa cells in this imaging study were treated with biocompatible concentrations of RGQDs (1.5 mg/mL) and HGQDs (1.7 mg/mL) for 20 h. This treatment time was chosen because it is close to the maximum cell internalization timeline of similar GQD structures [43,44,60]. Visible confocal fluorescence microscopy imaging suggests the internalization of both RGQDs and HGQDs showing their green 535 nm fluorescence within HeLa cells with lamp excitation at 460 ± 20 nm (Figure 6a,d). This result was further confirmed by the hyperspectral NIR imaging with an 808 nm laser excitation (Figure 6b,e). A 1080 nm bandpass setting of the hyperspectral imager was utilized to ensure that NIR fluorescence was observed only from the GQDS. While not entirely overlapping with visible, NIR emission provides a background-free assessment of GQD location and internalization. Hyperspectral microscopy-generated spectra taken from the regions of GQD fluorescence within the cells (Figure 6c,f) appeared to be similar to bulk GQD NIR fluorescence features validating the intracellular presence of GQDs (Figure 2b).

NIR emission can serve as a complementary bioimaging modality for a variety of photothermal treatment applications, including real-time image-guided ablation of tumors [83,91,92,93]. Tumor-targeted treatment can aid in locating tumors and further focus photothermal treatment to their location. Moreover, with their hydrophobic graphitic platform, an abundance of functionalizable oxygen groups, and a broad choice of potential precursor materials for top-down and bottom-up synthesis, GQDs can be nano-engineered to carry drug, gene, or targeting agents that can complement or enhance their PTT therapeutic effect [47,61,94,95,96]. Furthermore, in order to improve the therapeutic outcomes as well as enhance the platform’s biocompatibility, GQDs can also be incorporated into nanocomposite hydrogels [97,98]. Such future prospects coupled with successful PTT capabilities demonstrated in this work give HGQDs and RGQDs a substantial advantage over many other PTT nanomaterial agents in multimodal theragnostic applications.

## 4. Conclusions

In this work, graphene quantum dots, which were biocompatible up to 1.5–1.7 mg/mL, few nanometer-sized, water-soluble, were synthesized and tested as photothermal and bioimaging agents in vitro. Simple and scalable top-down and bottom-up approaches were utilized to produce RGQDs and HGQDs from reduced graphene oxide and hyaluronic acid precursor materials. Photothermal conversion capacity as well as PTT cancer cell treatment capabilities of these GQDs were evaluated with 0.9 W/cm^2^ 808 nm laser irradiation in the first NIR water absorption window. Aqueous solution experiments at 37 °C ambient temperature demonstrated laser-induced heating of RGQD and HGQD suspensions up to 47 and 46 °C, respectively, after 10–15 min of irradiation with no apparent heating of water controls. Both GQD types had remarkable photothermal stability, demonstrating similar heating after three consecutive irradiation cycles. A new high-throughput irradiation/measurement system based on the retrofitted 3D printer was developed to conduct multiple in vitro PPT studies in the same 96-well plate. This device, equipped with a laser and thermocouple for in-well irradiation and temperature measurements, was programmed to test the PTT effect of different GQD samples at 1–10 min irradiation times in HeLa cells. The effects of such PTT on cancer cell viability were evaluated in the same well plate via the MTT assay. As compared to the HeLa control experiencing only minor heating and/or decrease in cell viability upon irradiation, HeLa cells treated with RGQDs and HGQDs demonstrated substantial photothermal heating to 43.5 and 54.5 °C together with the decrease in cancer cell viability down to 38.2% and 22.9%, respectively. Visible and near-infrared GQD fluorescence was successfully utilized in this work for imaging both HGQDs and RGQDs in HeLa cells to test their image tracking capabilities and verify their successful internalization. Due to its greater penetration depth, NIR fluorescence offers a broader range of potential in vivo/ex vivo applications for GQDs as dual-mode imaging/photothermal platforms. Top-down- and bottom-up-synthesized GQDs tested in this work, therefore, show a significant potential as biocompatible image-guided photothermal agents, while their prospective drug delivery capabilities can further extend GQD applications into highly desired drug/PTT combination therapeutic approaches.

## Figures and Tables

**Figure 1 nanomaterials-13-00805-f001:**
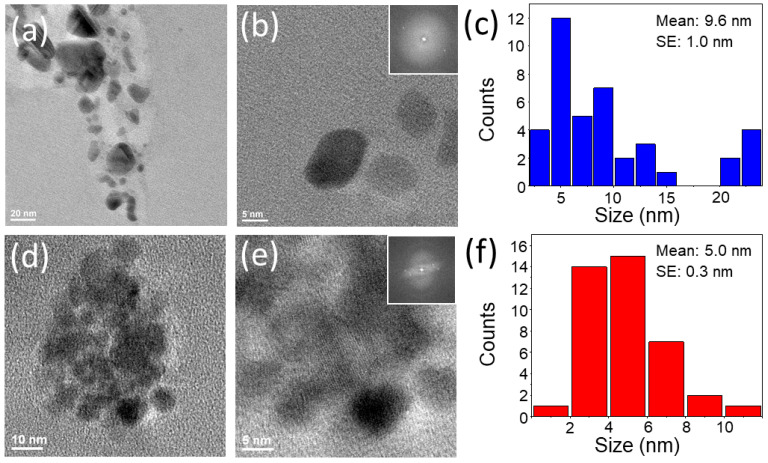
TEM analysis of RGQDs (**a**–**c**) and HGQDs (**d**–**f**) including representative TEM images (**a**,**d**), HRTEM scans with FFT images in the inset (**b**,**e**), and TEM-derived size distributions (**c**,**f**).

**Figure 2 nanomaterials-13-00805-f002:**
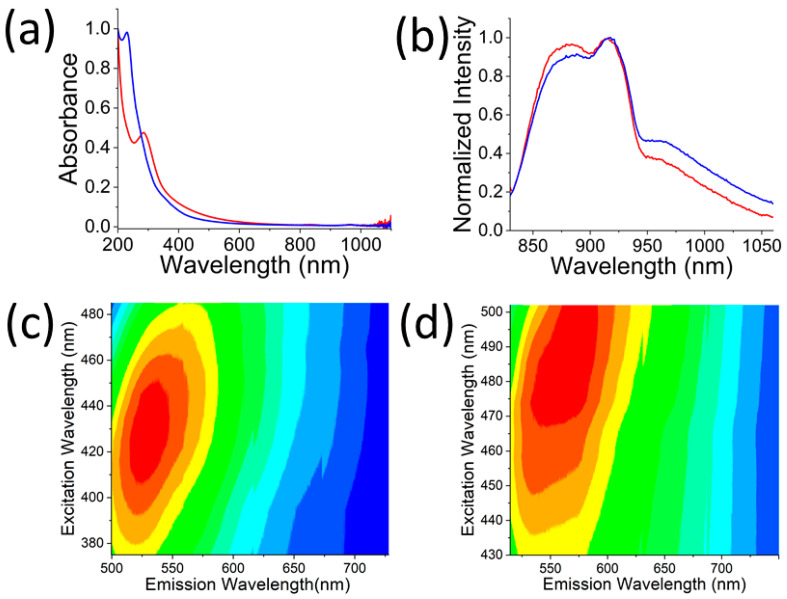
(**a**) Absorption spectra of HGQDs (red) and RGQDs (blue). (**b**) NIR fluorescence spectra of HGQDs (red), and RGQDs (blue) excited with 808 nm laser. Visible excitation/emission maps of (**c**) HGQDs and (**d**) RGQDs.

**Figure 3 nanomaterials-13-00805-f003:**
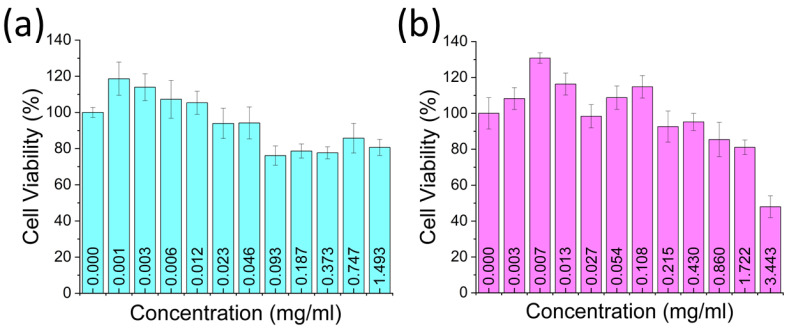
MTT assay-based viability of HeLa cells treated with different concentrations of (**a**) RGQDs and (**b**) HGQDs.

**Figure 4 nanomaterials-13-00805-f004:**
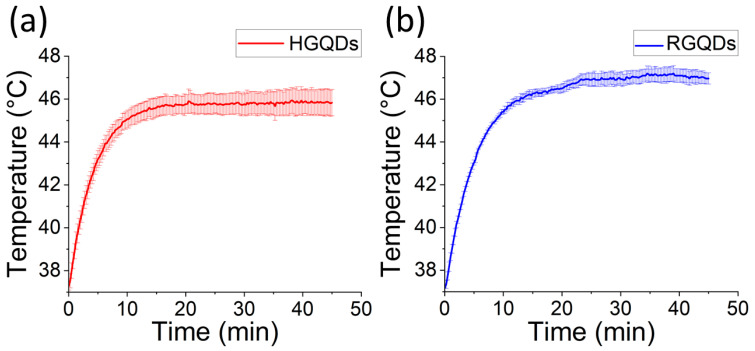
Temperature of (**a**) HGQD (1.7 mg/mL) and (**b**) RGQD (1.5 mg/mL) suspensions after 0.9 W/cm^2^ 808 nm laser irradiation for up to 47 min.

**Figure 5 nanomaterials-13-00805-f005:**
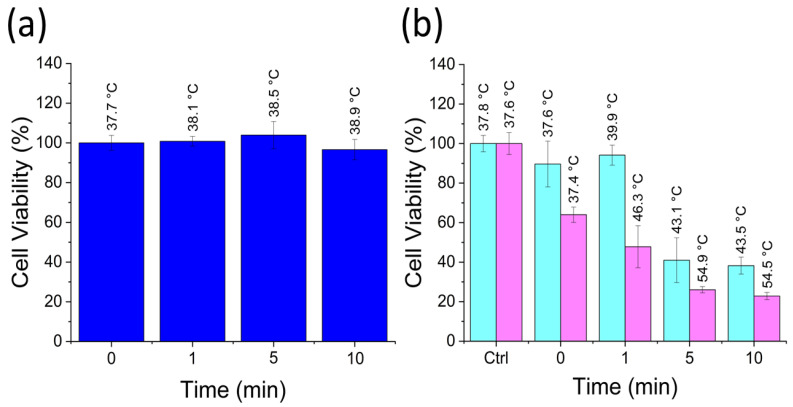
MTT assay-based viability of HeLa cells irradiated with an 808 nm (0.9 W/cm^2^) laser for 0, 1, 5, and 10 min. (**a**) HeLa nontreatment control. (**b**) HeLa cells treated with RGQDs at 1.5 mg/mL (cyan) and HGQDs at 1.7 mg/mL (pink).

**Figure 6 nanomaterials-13-00805-f006:**
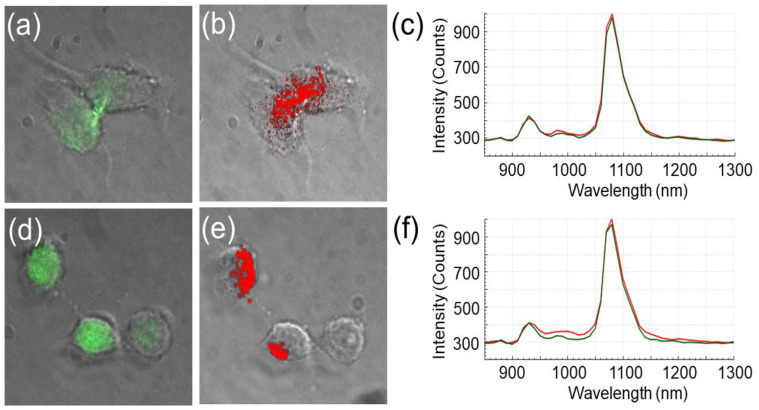
Fluorescence of RGQDs in the visible (**a**) and NIR (**b**) in HeLa cells. Fluorescence of HGQDs in the visible (**d**) and NIR (**e**) in HeLa cells. Spectra of RGQDs (**c**) and HGQDs (**f**) collected with NIR hyperspectral microscopy within HeLa cells (red) and background (green).

## Data Availability

The data presented in this study are available on request from the corresponding author. The data are not publicly available due to privacy.

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
