# Peer review of "Automated Approach to In Vitro Image-Guided Photothermal Therapy with Top-Down and Bottom-Up-Synthesized Graphene Quantum Dots"

_nanomaterials, 2023, doi:10.3390/nano13050805_

Round 1

Reviewer 1 Report

The paper deals out an investigation of phothermal properties of graphene quantum dots on Hela cells by heating them up to 54.5 °C. The paper is well written and well organized. The critical question of toxicity of graphene quantum dots is carefully considered. Although further studies will be needed to corroborate the highlighted results, the findings seems to be convincing and the graphene quantum dots could gain high interest in the community of photothermal therapists, in turn , as a bioimaging agents in vitro.

I suggest only few minor changes:

for a general discussion on photothermal approaches, I suggest to cite: M. D'Acunto, Detection of Intracellular Gold Nanoparticles. An overview, Materials 2018, 11, 882.

acronyms in the abstract should be avoided or introduced, such as, for example, HGQDs and RGQDs

Reviewer 2 Report

In this work, the authors used several GQD structures, including RGQD derived from reduced graphene oxide by top-down oxidation and HGQD synthesized from molecular hyaluronic acid hydrothermal bottom-up to test biocompatibility. These GQDs possess substantial NIR absorption and fluorescence throughout the visible and NIR beneficial for in vivo imaging, while being biocompatible at up to 1.7 mg/mL concentrations. In this study, HGQDs and RGQDs facilitate the heating of HeLa cancer cells up to 54.5 °C, leading to drastic inhibition of cell viability from over 80% down to 22.9%. GQD fluorescence in the visible and NIR traces their successful internalization into HeLa cells maximized at 20 hrs, suggesting both, the extracellular and intracellular photothermal treatment capabilities. The combination of the photothermal and imaging modalities tested in vitro makes GQDs developed in this work prospective agents for cancer theragnostics. I believe that publication of the manuscript may be considered only after the following issues have been resolved.

1.       In order to better highlight the advantages of this work, the author needs to provide a table to compare related work.

2.       What is the physical mechanism of the excellent performance of GQDs prepared in this work? Because I saw that there was no good breakthrough in the preparation method, and all of them were based on the existing methods.

3.       The summary needs to be simplified, and the background involved is suggested to be deleted by the author. After all, the introduction will introduce the relevant background in detail.

4.       The introduction can be improved. The articles related to some applications of graphene materials should be added such as Sensors 2022, 22, 6483; ACS Sustain. Chem. Eng. 2015, 3, 1677–1685; Diamond & Related Materials 128 (2022) 109273; Talanta 2015, 134, 435–442.

5.       Please check the grammar and spelling mistakes of the whole manuscript.

Reviewer 3 Report

1.      Please define each abbreviation used in the text when it first appears, including in the abstract; in addition, consider using fewer abbreviations in order to make the text easily accessible to a broad audience.

2.      Please summarize different types of graphene-based for quantum dot. What are the main differences for Photo-thermal therapy of top-down and bottom-up-synthesized graphene quantum dots in this study?

3.      How about the colloidal stabilityof top-down and bottom-up-synthesized graphene quantum dots?

4.      Please comment on the potentials of GQDs-based nanoplatforms for clinic applications. Whether it can be eventually commercially used for clinic?

5.      Some closely related work on the biomedical applications of other related bionanomaterials should have a comparison with them in the discussion part: https://doi.org/10.1016/j.colsurfb.2021.112025, https://doi.org/10.1016/j.colsurfb.2020.111243.

Round 2

Reviewer 2 Report

Accept in present form.